# *Hi Guys* or *Hi Folks*? Benchmarking Gender-Neutral Machine Translation with the GeNTE Corpus

**Andrea Piergentili,**[1,2]* **Beatrice Savoldi,**[2]*
**Dennis Fucci,**[1,2] **Matteo Negri**[2]**, Luisa Bentivogli**[2]
[1]University of Trento
[2]Fondazione Bruno Kessler
{apiergentili,bsavoldi,dfucci,negri,bentivo}@fbk.eu

## Abstract

Gender inequality is embedded in our communication practices and perpetuated in translation technologies. This becomes particularly apparent when translating into grammatical gender languages, where machine translation (MT) often defaults to masculine and stereotypical representations by making undue binary gender assumptions. Our work addresses the rising demand for inclusive language by focusing head-on on gender-neutral translation from English to Italian. We start from the essentials: proposing a dedicated benchmark and exploring automated evaluation methods. First, we introduce GeNTE, a natural, bilingual test set for gender-neutral translation, whose creation was informed by a survey on the perception and use of neutral language. Based on GeNTE, we then overview existing reference-based evaluation approaches, highlight their limits, and propose a reference-free method more suitable to assess gender-neutral translation.

## 1 Introduction

Societal gender asymmetries and inequalities are reflected and perpetuated through language (Stahlberg et al., 2007; Menegatti and Rubini, 2017). Such awareness has grown also within the Natural Language Processing (NLP) field (Blodgett et al., 2020), where extensive research has highlighted how several applications suffer from gender bias (Sun et al., 2019; Sheng et al., 2021). As also noted by MT users themselves (Olson, 2018; Dev et al., 2021), among these applications are translation systems used at large scale, which pose the concrete risk of misrepresenting gender minorities by over-producing masculine forms, while reinforcing binary gendered expectations and stereotypes (Savoldi et al., 2021; Lardelli and Gromann, 2022).

To foster greater inclusivity and break free from the constraints of masculine/feminine language,

---
*The authors contributed equally.

neutral strategies have emerged and are increasingly adopted in academia (APA, 2020), institutions (Höglund and Flinkfeldt, 2023), and industry alike (Langston, 2020). These strategies aim to overcome marked forms that treat the masculine gender as the conceptually generic, default human prototype (e.g., *humankind* vs. *mankind*) (Silveira, 1980; Bailey et al., 2022). Thus, they challenge gender norms and embrace all gender identities by avoiding gendered terms when unnecessary (e.g. *chair* vs. *chairman/chairwoman*) (Hord, 2016).

English, being at the forefront of inclusive language changes and with its limited gendered grammar (Ackerman, 2019), has faced fewer obstacles in adapting to neutral forms, which have already been modeled into monolingual generative tasks (Sun et al., 2021; Vanmassenhove et al., 2021). As recently underscored by Amrhein et al. (2023), however, the resources and approaches made available for English are not portable to grammatical gender languages. Such need for dedicated efforts is exemplified in Italian, where neutral solutions must navigate the extensive encoding of masculine/feminine marking (e.g. *the doctors are qualified* → it: *i/le dottori/esse sono qualificati/e*) through synonymy or more complex rephrasing (Papadimoulis, 2018) (e.g. → *il personale medico* [the medical staff]). While indeed more challenging, pursuing inclusivity in Italian is relevant exactly because sexist attitudes are more visible and impactful in grammatical gender languages (Wasserman and Weseley, 2009). Nonetheless, the implementation of neutral language in MT remains to date a basically uncharted territory, despite the desirability of neutral outputs under several circumstances where gender is ambiguous or irrelevant.

In light of the above, by focusing on English→Italian as an exemplary and representative translation pair and direction, we hereby lay the groundwork toward gender-neutral MT. Starting from a survey aimed to understand the challenges

of neutral translation in cross-lingual settings, we provide the necessary tools and resources to foster research on the topic by estimating gender neutral translation in MT. Hence, our main contributions are: **(1)** A study on the feasibility of neutral translation, by surveying the potential trade-off among fluency, adequacy, and neutrality; **(2)** The creation of GeNTE,[1] the first natural, parallel corpus designed to test MT systems' ability to generate neutral translations; **(3)** A comprehensive analysis of the (un)suitability of existing automatic metrics to evaluate neutral translation. As an inherent benchmark component, we indicate an alternative solution capable to better assess the task.

We make the GeNTE dataset freely available at `https://mt.fbk.eu/gente/` and release the evaluation code under Apache License 2.0 at `https://github.com/hlt-mt/fbk-NEUTR-evAL`.

## 2 Background

Emerging research has highlighted the importance of reshaping gender in NLP technologies in a more inclusive manner (Dev et al., 2021), also through the representation of non-binary identities and language (Wagner and Zarrieß, 2022; Lauscher et al., 2022; Ovalle et al., 2023). Foundational works in this area have included several applications, such as coreference resolution systems (Cao and Daumé III, 2020; Brandl et al., 2022), intra-lingual fair rewriters (Amrhein et al., 2023), and automatic classification of gender-neutral text (Attanasio et al., 2021).

In MT, the research agenda has mainly focused on the improvement of masculine/feminine gender translation. Along this line, different mitigation methods have been devised to ensure that unambiguous gendered referents (e.g. **he/she** is a doctor) are properly resolved in the target language (Costa-jussà and de Jorge, 2020; Choubey et al., 2021; Saunders et al., 2022). These methods are often tested on synthetic template-based datasets such as WinoMT (Stanovsky et al., 2019) or Simple-GEN (Renduchintala and Williams, 2022). As also stressed by Saunders and Olsen (2023), however, in realistic scenarios MT systems are also confronted with ambiguous input sentences that do not convey any gender distinction (e.g., en: *I called the **doctor***). Nonetheless, to date the resources and solutions envisioned for resolving such cases into grammatical

gender languages like Arabic (Alhafni et al., 2022), Italian (Vanmassenhove and Monti, 2021), Spanish, or French (Rarrick et al., 2023) entail offering two possible translation outputs, still constrained to binary gender forms (e.g., it: *Ho chiamato **il dottore*** MASC vs. ***la dottoressa*** FEM).[2]

As an exception within the current MT landscape, Cho et al. (2019) and Ghosh and Caliskan (2023) investigate the preservation of gender-ambiguous pronouns for Korean/Bengali→English. Since English can already boast the well-established neutral pronoun *they*, their study does not face the additional challenges of preserving such unmarked vagueness into grammatical gender languages. Such challenges are exemplified by Saunders et al. (2020), who created parallel test and fine-tuning data to develop MT systems able to generate non-binary translations for English→German/Spanish. However, their target sentences are artificial – created by replacing gendered morphemes and articles with synthetic placeholders – thus serving only as a proof-of-concept. To the best of our knowledge, Piergentili et al. (2023) are the first to advocate the use of target gender-neutral rephrasings and synonyms as a viable paradigm toward more inclusive MT when gender is unknown or simply irrelevant. Despite this call to action, no concrete steps have been taken yet to actually facilitate research in this direction, not even toward suitable benchmarks to recognize the neutral forms occasionally generated by current systems (Savoldi et al., 2022).

In light of the above, the path toward gender-neutral translation in MT is bottlenecked by the lack of dedicated datasets and automated evaluations. Here, we fill this gap so to guide and allow research on this novel topic. To this aim, we start in §3 by first ensuring that gender-neutral language can enable acceptable translations, not being perceived as inappropriate or intrusive.

## 3 Surveying Gender-Neutral Translation

Neutralization is a form of linguistic gender inclusivity that relies on the retooling of established forms and grammar (Gabriel et al., 2018). According to the review of several gender-inclusive public guidelines by Piergentili et al. (2023), these can range from *i)* simple word changes, like omissions or article/noun replacements with epicene alterna-

---

[1] **Ge**nder-**N**eutral **T**ranslation **E**valuation. In Italian, *gente* means *folks*, a term used for inclusive greetings in lieu of "*guys*".

[2] Such double-outputs are currently offered for short, ambiguous queries also by Google Translate and Bing.

| Questionnaire Example sentences | Eq. | NT | GT |
|---|---|---|---|
| **tot. responses** | 36.5 | 42.5 | 21 |
| A. *Some metals may be toxic to* **man**.
**GT** Certi metalli possono essere tossici per **l'uomo**.
**NT** Certi metalli [...] per **gli esseri umani** [humans]. | 39.6 | 50 | 10.4 |
| B. *Does* **anyone** *wish to speak against the proposal?*
**GT** **Qualcuno** desidera intervenire contro la proposta?
**NT** Ci sono interventi[Are there speeches] contro [...] | 54.7 | 31.6 | 13.7 |
| C. *Indonesia is dealing with one million* **refugees**...
**GT** L'Indonesia ha un milione di **profughi**...
**NT** L'Indonesia ha un milione di **esuli**[exiles]... | 40 | 23.2 | 36.8 |

Table 1: Questionnaire example of English sentences with translation alternatives. For each example, participants responses – *GT and NT are equivalent, NT is preferrable, GT is preferrable* – are shown (percentage).

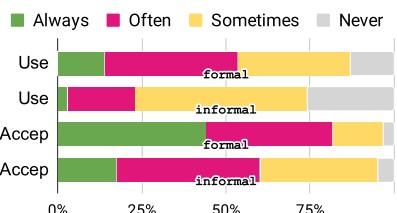

Figure 1: Frequency of use and acceptance of neutral language in *formal* vs. *informal* communication.

tives (e.g. **il maestro** vs. *l'insegnante*[3]), to *ii)* more complex reformulations, which might involve altering the sentence structure (e.g. **i miei colleghi** *vs. le persone con cui lavoro).*[4] As such, to ensure neutrality, these solutions might have an effect in terms of brevity or perceived fluency.

While widespread in monolingual, institutional contexts (Papadimoulis, 2018), the use of neutral forms in cross-lingual settings requires to weigh additional non-negligible factors. First, translations are bounded to a source text, whose meaning must be properly rendered in the target language. Thus, more creative reformulations might collide with this instrinc constraint. Also, it might be not always clear-cut when neutral translations ought to be performed. This is the case of masculine generics in the source language (e.g. *All* **firemen**): while they do not neatly fall under the idea of "ambiguous" input, their propagation to a target language clashes with the goal of inclusive MT itself.

These issues stand unaddressed: the study by Lardelli and Gromann (2023) represents the only empirical investigation on the feasibility of gender-neutral translation, but it is concerned with the cognitive effort that its realization poses to post-editors. Therefore, to better understand the implications of gender-neutral translation for a wider range of stakeholders, we carried out a preliminary analysis on English→Italian by surveying the opinions of potential MT end-users.

**Questionnaire.** Our survey was structured into two main parts. In part *(i)*, we indirectly assessed linguistic acceptability: given a source English sentence paired with both a gendered (GT) and a neutral (NT) translation, we asked participants to indicate whether they had a preference or found

them to be equivalent[5] (see Table 1). Then, in part *(ii)* we asked direct questions to gauge participants' use and attitude toward gender-neutral language. The questionnaire was distributed online and received 98 responses by eligible participants. While all details are provided in Appendix A, here we summarize our main insights.

First, the linguistic acceptability of gender-neutral translations was positively judged, perhaps at the higher rate than our own expectations. In fact, overall results indicate that in the majority of cases the NT was deemed preferable (42.5%) rather than equivalent to the GT (36.5%), where only a minority favoured gendered translations (21%). Possibly, such trend is explained in light of participants' ideological preference for inclusive language rather than by purely linguistic factors. As shown in Table 1, this seems to be confirmed by disaggregating responses for each translated example sentence, where we can extract more qualitative insights. Indeed, example A attests one of the strongest preferences for NT, signalling a negative attitude toward the propagation of default, masculine forms in GT. Then, concerning the use of more of complex neutral rephrasings (example B), we found that slightly longer and sentence-altering neutralization strategies were still considered largely acceptable. Instead, literal NT with limited changes, which however sacrificed more the source meaning or altered its tone (example C), were comparatively penalized. This trend was also confirmed in part *(ii)* of the survey (see Appendix A). Finally, as shown in Figure 1, participants' responses to direct questions attest that neutralization strategies are accepted/used differently depending on the speech situation, with a preference for their use in formal communicative situations. Having confirmed the feasibility and overall acceptability of neutral translations, we embed the gathered qualitative considerations in the design of the GeNTE corpus (§4).

---

[3]en: "the teacher".

[4]en: "my colleagues" vs. "the people I work with".

[5]i.e. equally adequate and fluent.

## 4 The GeNTE corpus

GeNTE is the first test set designed to evaluate MT models' ability to perform gender-neutral translations, but only under desirable circumstances. In fact, when referents' gender is unknown or irrelevant, undue gender inferences should not be made and translation should be neutral. However, neutralization should not be always enforced; for instance, when a referent's gender is relevant and known, MT should not over-generalize to neutral translations. The corpus hence consists of 1,500 English-Italian parallel sentences with mentions to human referents that equally represent two translation scenarios: **1)** Set-N, featuring gender-ambiguous source sentences that require to be neutrally rendered in translation; **2)** Set-G, featuring gender-unambiguous source sentences, which shall be properly rendered with gendered (masculine or feminine) forms in translation. Altogether, these sets allow to benchmark whether systems are able to perform gender-neutral translation, and if they do so when appropriate.

We build GeNTE on naturally occurring instances of both scenarios retrieved from Europarl (Koehn, 2005). Besides being a widely popular and high-quality MT resource, we chose this corpus inasmuch it represents formal communicative situations from the administrative/institutional domain. Accordingly, it reflects the context for which gender-neutral forms are traditionally intended, also in line with the stakeholders' preference highlighted in §3. Also, as examined by Saunders (2022), Europarl exhibits a large amount of gender-ambiguous cases that – although translated with gendered forms in the original references of the corpus – lend themselves as suitable candidates for neutralization. As explained in the forthcoming paragraphs (§4.2), for each of these original Europarl gendered target sentences, we create an additional gender-neutral reference translation.

### 4.1 Data selection and annotation

**Data extraction.** To retrieve Europarl[6] segments representing our two translation scenarios of interest, we crafted regular expressions to: *i)* identify source sentences containing mentions to human referents, *ii)* maximize the variability of linguistic phenomena included in the corpus, and *iii)* ensure a balanced distribution of both unambiguous and ambiguous gender translation cases. To this aim, we

targeted Set-G segments by matching source English sentences that contained explicit gender cues, e.g. lexically gendered words (*sister*, *woman*), titles (*Mr*, *Mrs*) and marked pronouns (*him*, *her*). Set-N, instead, was populated by matching several word classes that do not convey any gender distinction in English (e.g. *you*, *citizens*, *went*), but typically correspond to masculine/feminine expressions in the target language. Also, we searched for masculine terms used generically, such as *man* and its derived compounds (e.g., *chairman*, *layman*). In fact, masculine generics are unreliable gender cues and, following the survey findings (§3), should not be propagated in MT.

**Sentence editing.** On the collected material, a first intervention was carried out to streamline the evaluation of gender-neutral translation. In fact, some of the source sentences contained mentions of multiple referents, which required the combination of different forms in translation (i.e. neut/masc/fem). In those cases, the parallel sentences were manually edited so as to ensure that they only include referents that require the same type of (either neutral or gendered) forms. In this way, each sentence pair can be handled as a whole coherent unit, thus avoiding the complexities of evaluating intricate combinations of phenomena. To ensure a balanced distribution of instances from both Set-N and Set-G, a second intervention was required to compensate for the under-representation of unambiguous cases.[7] Although these edits slightly reduce the naturalness of the data, they allow for a simpler and sound evaluation, crucial to shed light on a complex task such as gender-neutral MT. Instead, other edits were made to enhance the quality of the corpus; all of them are reported in Appendix §B.1. Once the editing phase was concluded, all sentence pairs were annotated as N in Set-N, and as F or M in Set-G. In the annotation process, it was verified that the initial pool of – automatically extracted – candidate sentences were correctly assigned to Set-N and Set-G by accounting for the sentence context. In this way, we could differentiate between the use of gendered words as either masculine generics (e.g. *It is up to an accused employer to prove **his** innocence* – identified as N) or as informative of a referent's gender (e.g. *I would like to thank Commissioner Byrne for **his** cooperation.* – identified as G).

---

[6] https://www.statmt.org/europarl/archives.html

[7] Such lack confirms the vast representation of generic and unknown referents in Europarl, as found in Saunders (2022).

| | | SRC | I, along with **all my colleagues**, wish to welcome this [...] |
|---|---|---|---|
| *i* – N | | REF-G | Insieme a **tutti i miei colleghi**, desidero esprimere il mio compiacimento per questa [...] |
| | | REF-N 1 | Insieme a **agli altri membri**[other members], desidero esprimere il mio compiacimento per questa [...] |
| | | REF-N 2 | Insieme a **ogni collega**[each colleague], desidero esprimere il mio compiacimento per questa [...] |
| | | REF-N 3 | Insieme a **tutte le persone con cui lavoro**[all the persons with whom I work], desidero esprimere il mio compiacimento per questa [...] |
| | | SRC | I welcome this excellent report **from my colleague** Mr Skinner. |
| *ii* – M | | REF-G | Valuto positivamente la relazione **del collega**, onorevole Skinner. |
| | | REF-N 1 | Valuto positivamente la relazione **dell'onorevole collega**[of the honorable colleague] Skinner. |
| | | REF-N 2 | Valuto positivamente la relazione **dell'onorevole collega** Skinner. |
| | | REF-N 3 | Valuto positivamente la relazione **dell'onorevole collega** Skinner. |
| | | SRC | Mrs Ana de Palacio Vallelersundi has **a** sister **who is a Commissioner** [...] |
| *iii* – F | | REF-G | **La** onorevole [...] ha **una sorella, la quale è una Comissaria** [...] |
| | | REF-N 1 | N.A. |
| | | REF-N 2 | L'onorevole [...] ha **uno stretto legame di parentela**[is closely related] con **un membro della Commissione**[a member of the Commission] |
| | | REF-N 3 | N.A. |

Table 2: Examples of entries in the COMMON-SET. REF-G indicates the gendered references, REF-N 1, 2, 3 indicate the neutralized references produced by Translator 1, 2, and 3 respectively. Words in **bold** are mentions of human referents; underlined words are linguistic cues informing about the referents's gender.

## 4.2 Creation of gender-neutral references

As a confirmation of the predominant use of gendered forms when translating into grammatical gender languages, it is worth remarking that almost all (97.2%) segments collected from Europarl have gendered references in Italian. Inspired by the design of natural (binary) gender bias benchmarks such as MuST-SHE (Bentivogli et al., 2020) and MT-GenEval (Currey et al., 2022), we thus created a second translation, so to allow for a reference-based contrastive evaluation of gender-neutral MT (see §5.2). To this aim, for each sentence pair, we created an additional Italian reference, which differs from the original one only in that it refers to the human entities with neutral expressions. This makes it possible to isolate gender-related linguistic elements as the only source of variation in the score of system outputs when evaluated against both the gendered and the neutral references. As neutralization is an open-ended task that entails a high degree of variability in the possible solutions, we wanted such variability accounted for in the neutral references. Therefore, their creation was assigned to three professional translators hired via a translation agency.[8] Each of them was assigned a different portion of the collected Italian references, to be post-edited so as to only replace gendered terms with neutral formulations. An expert linguist native speaker of Italian[9] prepared detailed instructions[10] drawing from existing guidelines for the institutional domain. After an initial training session, the linguist supported the translators throughout the process and finally checked all the neutralizations.

In Appendix B.2, we provide qualitative insights regarding revisions and supervision of the linguist.

**GeNTE COMMON-SET.** Whereas each translator was in charge of post-editing one given portion of the corpus, we also selected a common set of 200 references to be neutralized by all translators (henceforth referred to as the COMMON-SET); 100 were taken from the gendered set (COMMON-SET-G), and 100 from the neutral one (COMMON-SET-N). Thus, we obtained 200 source sentences, each paired with one (original) gendered reference and three (post-edited) neutral references. The creation of a COMMON-SET was primarily motivated by the goal of having a subset of the corpus that could be used to test the robustness of evaluation protocols and metrics across the three different neutral references (see §5.2). Orthogonally, it allowed us to measure linguistic variability among the neutral and gendered references (see Appendix B.3). Table 2 shows examples from the COMMON-SET, which confirm the findings of our preliminary survey (§3). Example *i* is representative of the variability that is inherent to the neutralization task. Example *ii*, instead shows a rare situation where all translators used the same neutralization device and produced an identical sentence. Finally, *iii* shows a gendered term, whose neutralization requires verbose periphrases that compromise the original text's fluency and style. This case was signaled as particularly difficult to neutralize: two translators out of three did not create a neutral reference. Overall, based on a manual analysis of the COMMON-SET, the translators produced three identical gender-neutral references in 13.57% of the cases, while an additional 8% of translations exhibited a high degree of similarity (e.g., the same neutral words are used, but in a different order).

---

[8]The cost paid to the agency was of 60 euros/hour, for a total of 14 hours of work for each translator.

[9]The linguist is one of the authors of the paper.

[10]Released together with the GeNTE corpus.

|  | GENTE | | | | | | |
|---|---|---|---|---|---|---|---|
|  | Source | | REF-G | | REF-N | | |
|  | # sentences | Avg length | # sentences | Avg length | # sentences | Avg length | # G-words |
| Set-N | 750 | 25.67 | 750 | 24.66 | 750 | 26.95 | 1,972 |
| Set-G | 750 | 26.51 | 750 | 25.26 | 750 | 26.55 | 2,148 |
|  | COMMON-SET | | | | | | |
|  | Source | | REF-G | | REF-N | | |
|  | # sentences | Avg length | # sentences | Avg length | # sentences | Avg length | # G-words |
| Set-N | 100 | 27.45 | 100 | 27.00 | 300 | 28.87 | 300 |
| Set-G | 100 | 26.99 | 100 | 26.34 | 300 | 27.57 | 299 |

Table 3: Corpus statistics for GENTE and its subset COMMON-SET. Both sets requiring gendered translations (Set-G) are equally balanced between F and M sentences. Average lengths are calculated ignoring punctuation. In the last column, we provide the number of gendered words in the REF-Gs that had to be neutralized in the REF-Ns.

These statistics are positive: they show that the GeNTE COMMON-SET exhibits a good level of variability ($\sim$79%), which is desirable to test open-ended generation tasks like MT. Also – and especially in light of the fact that the translators worked independently – the $\sim$21% of identical/similar neutralizations suggests that neutralizing translation is a challenging but feasible task.

To conclude, relevant statistics for GeNTE and its COMMON-SET are provided in Table 3.

## 5 Gender-neutral Evaluation Protocols

We complement our benchmark creation effort with a study on the possible approaches for using GeNTE to conduct automated evaluations of neutral MT. To this aim, we first define sound test-bed conditions (§5.1). On this basis, we then experiment with a contrastive, reference-based protocol to inspect the effectiveness of standard MT metrics to assess neutral translation (§5.2). Then, to overcome the limitations encountered with the reference-based approach, we implement a reference-free protocol (§5.3), which shows promise in advancing the task's evaluation.

### 5.1 Test-bed

To ensure a sound comparison between different automatic evaluation protocols, we built a test-bed based on the GeNTE COMMON-SET (§4.2, Table 3). Our test-bed includes relevant instances in relation to our task, namely gendered and gender-neutral automatic translations in equal proportion. On this basis, the analyzed evaluation approaches can be compared in their ability to reward systems that generate neutralized outputs only when due.

The automatic Italian translations of the COMMON-SET sources were generated with two leading commercial MT systems: Amazon Translate[11] and DeepL.[12] However, a manual inspection showed an almost complete lack of representation of gender-neutral translations in the outputs: gendered translations were generated for all but one of the COMMON-SET-N inputs.[13] This result revealed the unsuitability of such outputs for investigating the automated evaluation of neutral translation itself. Accordingly, to obtain neutral (MT-like) outputs to be included in the test-bed, we resorted to manually post-editing the 100 COMMON-SET-N translations generated with undue gender assignments. To do so, we leveraged our manually-created neutral references (§4.2): we substituted the neutral forms produced by the three professional translators to the gendered forms in the MT outputs, so as to make them neutral without altering the rest of the sentence.[14] For each system, we thus obtained three sets of neutral output sentences (one per translator), so to account for the robustness of different evaluation methods to the linguistic variability expressed in the inventory of neutralization strategies potentially applicable by humans and machines.

### 5.2 Reference-based Evaluation

#### 5.2.1 Setting

In this evaluation protocol we aim to verify whether common reference-based MT metrics can be effectively used to identify gendered and neutral translations. The protocol is based on the idea that if a system generates a gendered translation, its output will be rewarded when evaluated against a gendered reference and penalized when evaluated

---

[11] https://aws.amazon.com/translate/

[12] https://www.deepl.com/en/translator

[13] This provides a glimpse into the shortcomings of inclusivity within the current MT landscape.

[14] On average, 12% of the words present in the systems' output were substituted through post-editing, thus these edits have a minimal and circumscribed impact that does not alter the original output sentence.

| Metric | COMMON-SET-G | | | | | | COMMON-SET-N | | | | | |
|---|---|---|---|---|---|---|---|---|---|---|---|---|
| | DeepL | | | Amazon | | | DeepL | | | Amazon | | |
| | REF-G | REF-N | Δ% | REF-G | REF-N | Δ% | REF-N | REF-G | Δ% | REF-N | REF-G | Δ% |
| **BLEU** | 34.95 | 27.97 | **19.98** | 35.20 | 28.12 | **20.11** | 24.91 | 22.82 | **8.39** | 24.44 | 22.44 | **8.19** |
| **chrF** | 64.18 | 58.52 | **8.82** | 64.01 | 58.32 | **8.90** | 55.49 | 55.81 | -0.59 | 55.54 | 55.76 | -0.40 |
| **TER ↓** | 52.18 | 59.68 | **14.38** | 53.54 | 61.35 | **14.59** | 66.52 | 70.99 | **6.73** | 66.68 | 71.32 | **6.97** |
| **METEOR** | 62.10 | 54.26 | **12.63** | 60.90 | 52.99 | **13.00** | 48.34 | 47.37 | **2.00** | 47.79 | 46.90 | **1.86** |
| **BERTscore** | 88.34 | 86.16 | **2.47** | 88.00 | 85.79 | **2.52** | 84.25 | 84.36 | -0.13 | 84.13 | 84.20 | -0.08 |
| **COMET** | 87.89 | 86.08 | **2.06** | 87.36 | 85.50 | **2.13** | 84.89 | 85.06 | -0.20 | 84.69 | 84.92 | -0.27 |
| **BLEURT** | 80.50 | 77.12 | **4.10** | 79.67 | 76.36 | **4.15** | 76.30 | 76.79 | -0.64 | 75.36 | 75.80 | -0.59 |

Table 4: Corpus-level scores for DeepL and Amazon Translate, and percentage **gains** (Δ%, with sign changed for TER) with respect to the correct references. COMMON-SET-G: the original MT output is evaluated against each of the three available references, resulting scores are averaged. COMMON-SET-N: each of the three edited MT outputs is evaluated against the two references not used to neutralize it, all resulting scores are averaged.

against a neutral one. On the contrary, if a system produces a neutral translation, this is expected to be rewarded when compared to a neutral reference and penalized when compared to a gendered one.

**Contrastive Protocol.** Given a system output and a reference-based metric, we compute corpus-level scores against both the gendered and the neutral references provided in COMMON-SET. Then, for COMMON-SET-G the metric is effective if the scores are higher when computed against the gendered translations than the neutral ones; vice versa for COMMON-SET-N.

**Metrics.** We study the effectiveness of a set of widely used metrics. These can be categorized as *i)* n-gram overlap metrics: BLEU[15] (Papineni et al., 2002), chrF (Popović, 2015), TER (Snover et al., 2006), and METEOR (Banerjee and Lavie, 2005), which are sensitive to surface form differences between outputs and references (Glushkova et al., 2023); *ii)* neural model-based metrics: BERTScore (Zhang et al., 2020), BLEURT (Sellam et al., 2020), and COMET (Rei et al., 2020), which compare semantic representations based on the respective underlying models.

### 5.2.2 Results

Table 4 reports the results computed with each metric on our test-bed. First, results are consistent between DeepL and Amazon Translate. With respect to COMMON-SET-G, all the metrics correctly give higher scores for gendered references than for neutral ones: positive percentage differences thus indicate that the metrics correctly reward systems' gendered translations. However, in COMMON-SET-N there is a divergence in performance between n-gram overlap metrics and neural metrics. Only three metrics based on n-gram overlap – BLEU, TER, and METEOR – correctly

assign higher scores to systems evaluated against the neutral references.

These results show that, compared to neural metrics, n-gram overlap metrics appear more suitable for assessing gender-neutrality. The lower effectiveness of the neural metrics could be attributed to the lower frequency of neutral expressions in the training data of these models, leading to a lower probabilities assignment. Also, neural metrics are sensitive to semantic variations, but robust to surface lexical or morphological differences. Thus, since gender neutralizations preserve the essential semantics of their gendered equivalents, neural metrics are unable to properly frame their differences in this contrastive setting. This is evident in the consistently higher Δ percentages observed in COMMON-SET-G for n-gram overlap metrics (all above 8%) compared to the lower percentages obtained with neural metrics (all below 5%).

We further experimented with BLEU, TER, and METEOR to investigate their ability to provide more fine-grained evaluations. We thus tested them on the same data and the same contrastive principle, but at the sentence level (i.e. a neutral output sentence should obtain higher scores on the neutral reference compared to the gendered reference, and vice versa for a gendered output translation). In this way, the Δ obtained with the contrastive reference-based evaluation protocol that relies on BLEU, TER and METEOR can be used to categorize each sentence as either gendered or gender-neutral: higher BLEU on the neutral reference wrt. the gendered reference → output sentence classified as neutral; higher BLEU on the gendered reference wrt. the neutral reference → output sentence classified as gendered. By distinguishing the sentences belonging to COMMON-SET-N and COMMON-SET-G in advance, we can thus calculate an overall accuracy that represents the proportion of sentences correctly categorized. The results are

---

[15]BLEU|#:1|c:mixed|e:no|tok:13a|s:exp|v:2.3.1

presented in Table 5. For COMMON-SET-G, the performance is rather promising for all the three metrics, with accuracy scores always above 90% for the outputs of both Amazon Translate and DeepL. This is in line with the corpus-level results for the same set. Interestingly, though, for COMMON-SET-N the accuracy scores are very low, worse than or close to random choice for METEOR and BLEU. Only TER-based evaluations are higher (65.17% for Amazon Translate, 65.83% for DeepL).

In conclusion, through this closer inspection, we find that none of the n-gram overlap metrics is actually reliable for the evaluation of gender-neutral translation. This is possibly due to the fact that our reference-based evaluation approach, just like the metrics it is based on, is heavily dependent on the reference sentences. Outputs that deviate from the reference, even if they are acceptable translations, may therefore be penalized. This issue becomes particularly critical when evaluating gender-neutral translations, as periphrasis or synonymy are among the most common and accepted techniques used for achieving neutrality (§3). These strategies are inherently penalized by n-gram overlap metrics and do not seem to entail significant differences in meaning according to neural metrics, thus advocating for alternative, reference-free solutions.

### 5.3 Reference-free Evaluation

#### 5.3.1 Setting

Our reference-free protocol for the evaluation of gender-neutral translation explores a classification-based approach. We cast the problem as a binary task to classify if automatically-translated sentences are gendered or neutral. Implementing this procedure requires *i)* generating training data, and *ii)* training the classifier on the collected data. Then, results are computed on our test-bed in terms of classification accuracy.

**Synthetic Data Generation.** Confronted with the lack of Italian corpora featuring gender-neutral language, we resorted to synthetic data generation by prompting GPT (gpt-3.5-turbo). To do so, we devised a three-step approach that allowed for a more controlled generation procedure with reduced risk of noise (for full details, see Appendix C.1). First, similarly to Attanasio et al. (2021), we manually created 800 triplets of neutral, masculine, and feminine referents (e.g. the neighbours: *il vicinato - i vicini - le vicine*). Then, we used such triplets as seedwords to prompt GPT and generate

| Metric | DeepL | | | Amazon | | |
|---|---|---|---|---|---|---|
| | Set-G | Set-N | All | Set-G | Set-N | All |
| **BLEU** | 92.00 | 52.00 | 72.00 | 93.33 | 52.66 | 73.08 |
| **TER** | 90.33 | 65.83 | 78.08 | 91.67 | 65.17 | 78.42 |
| **METEOR** | **94.67** | 42.71 | 68.69 | **94.67** | 41.43 | 68.05 |
| **Classifier** | 91.00 | **88.67** | **89.83** | 87.00 | **87.33** | **87.17** |

Table 5: Accuracy scores for reference-based (BLEU, TER, and METEOR) and reference-free (classifier) evaluation protocols.

triplets of sentences, which only differ for the inserted (neut/masc/fem) seedword. This resulted in ∼60,000 sentences with a very low level of noise, but featuring a rather simple and repetitive syntactic structure. Therefore, we finally carried out a second generation round, prompting GPT to rewrite each triplet adding context to increase sentence variability and length. This led to a final synthetic corpus of ∼380,000 sentences, equally distributed across neut/masc/fem instances, and with varied structures to favor generalization.[16]

**Gender-Neutral Classifier.** To implement the classification model, we leveraged UmBERTo,[17] a Roberta-based language model (LM) fine-tuned on the Italian section of the web corpus OSCAR (Ortiz Suárez et al., 2019). In the survey by Tamburini (2020), UmBERTo was proven to be one of the best-performing LMs for Italian. Given a sequence of tokens, UmBERTo returns a contextualized vector for each token, including the special [CLS] token placed at the beginning of the sentence. As suggested by Devlin et al. (2019), we added a linear layer on top of the [CLS] vector to predict the neutral or gendered class. We trained the parameters of both the linear layer and UmBERTo on the classification task using the synthetic corpus labeled with neutral or gendered – for feminine and masculine – tags. Since this solution yielded the best results, our final classifier was trained in unbalanced data conditions, by making use of all synthetic gendered sentences (e.g. 1/3 fem and 1/3 masc) and all neutral sentences (1/3 neut). For complete details on the training setup see Appendix C.2.

#### 5.3.2 Results

Compared to the accuracy scores obtained via sentence-level contrastive evaluation based on BLEU, TER, and METEOR (first three rows of

---

[16]The synthetic corpus is released with the evaluation model.

[17]https://huggingface.co/Musixmatch/umberto-commoncrawl-cased-v1

Table 5), it is evident that the scores achieved by the trained classifier (row 4) are notably higher. These discrepancies primarily arise from the performance on the neutral outputs (Set-N), where the classifier outperforms the best n-gram overlap metric (TER) by a margin of up to 22.67 points for DeepL. However, for gendered outputs, the classifier demonstrates slightly lower results compared to the three reference-based metrics (except for DeepL, where it outperforms TER). As a result, for our reference-free approach, the gap between the scores obtained on gendered and neutral outputs is small (especially for Amazon Translate), attesting a balanced performance across the two classes.

All in all, the proposed reference-free evaluation protocol appears a promising evaluation method to accompany the release of GeNTE and favour its future utilization as a benchmark for gender-neutral MT. It proves to be a robust approach, capable of handling the linguistic variability associated with gender neutralization strategies, and overcoming the limitations of the reference-based approaches.

## 6 Conclusions

In this work, we investigated gender-neutral translation as a path for inclusive MT. To this aim, we focused on English→Italian, a pair that is highly representative of the challenges of implementing neutral forms into grammatical gender languages. As a novel area of inquiry, we started from the fundamentals. First, we conducted a survey on the acceptability of gender-neutral translation, which highlighted the openness of potential MT end-users, especially in formal communicative situations. Second, informed by the survey, we built GeNTE, the first natural benchmark for evaluating gender-neutral translation in MT. Third, we investigated the (un)suitability of existing automatic evaluation protocols to assess gender-neutral translation, and thus proposing an alternative, reference-free solution. Having taken the first steps toward gender-neutral MT, our resources and evaluation method are made available to foster and inform the future development of more inclusive MT.

## Limitations

Naturally, this work presents some limitations. In the paper we took the very first steps to enable evaluation and research on the task of gender-neutral translation for inclusive MT into grammatical gender languages. To do so, we provided data

(§4 & §5.3) and modeling (§5.3) for the specific English→Italian language pair. Thus, except for the GPT prompts written in English, the released GeNTE neutral references and the trained classifier cannot be directly used for other target languages. However, Italian was chosen as a highly representative example of the challenges faced in cross-lingual transfer from English. Accordingly, we believe that our design considerations, the methodology for the creation of GeNTE, as well as the presented evaluation protocols broadly apply to other target grammatical gender languages, too.

In the experiments, we relied on different closed-source models: Amazon Translate, DeepL and GPT (gpt-3.5-turbo). This has reproducibility consequences, since these models are regularly updated, thus potentially yielding future results that differ from those reported in this paper. Also, their access via API (paid in US dollars) might not be affordable for all institutions/researchers.

Finally, due to the inability of current MT models to generate gender-neutral translations, the output sentences used to test different MT metrics and evaluation protocols (§5) were partially post-edited. Indeed, this solution does not completely reflect standard evaluation practices conducted on fully MT generated output. However, this post-editing process only targeted gender-related aspects of the output sentences, thus still vastly preserving the MT generation and offering a controlled, realistic scenario. It should be recognized though that, by design, we enacted evaluation conditions where the MT models *succeeded* in generating the expected (either neutral or gendered) output translation. Instead, since the models' outputs did not exhibit cases where MT *failed* at generating the expected (gendered) translation, we could not test the robustness of our evaluation protocols for such a scenario.

## Ethics Statement

By addressing inclusivity in MT, this work presents an inherent ethical component. It builds from concerns toward the societal impact of widespread translation technologies that reflect and propagate discriminatory and exclusionary language. Concretely, by potentially feeding into existing stereotypes, reinforcing male-grounded visibility, and perpetuating the erasure of non-binary gender iden-

tities.[18]  Still, our work is not without risks either and thus warrants some ethical considerations. In particular, we propose the use of gender-neutralization strategies that avoid the use of unnecessary gendered terms via the retooling of established forms and grammars. These strategies can be considered as a form of Indirect Non-binary Language (INL) (Attig and López, 2020), which are intended – as we do in this paper – to *equally elicit* all gender identities in language and prevent misgendering by excluding any kind of gender assumption (Strengers et al., 2020). Instead, Direct Non-binary Language (Attig and López, 2020) – emerging via grassroots efforts and more predominately in online social medias (Lauscher et al., 2022) — resort to the creation of neologisms, neopronouns or even neomorphology to typically *enhance* the visibility of non-binary individuals.

In light of the above, several, concurring forms can serve different inclusive language needs (Comandini, 2021; Knisely, 2020). Thus, it should be stressed that the neutralization strategies incorporated in our MT work are not prescrively intended. Rather, they are orthogonal to other attempts and non-binary expressions for inclusive language (technologies) (Lauscher et al., 2023; Ginel and Theroine, 2022).

## Acknowledgements

This work is part of the project "Bias Mitigation and Gender Neutralization Techniques for Automatic Translation", which is financially supported by an Amazon Research Award AWS AI grant. Moreover, we acknowledge the support of the PNRR project FAIR - Future AI Research (PE00000013), under the NRRP MUR program funded by the NextGenerationEU.

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

## A  Gender-Neutral Translation Survey

The questionnaire was released online in April 2023. We distributed it via targeted emails and social media posts, with a request to share with relevant groups. Participation in this survey was voluntary, uncompensated and anonymous, as no identifying or personal information about the participant was collected. Also, participants were free to withdraw at any time without penalty or consequence. An anonymized version of the survey is accessible here: `https://forms.gle/YL76UeWbe4NWdCPPA`.

Note that we did not target individuals who use MT as a professional tool. Rather, generic stakeholders that might have used MT directly or indirectly (e.g. being offered translations of web pages). They were made aware that results of the questionnaire would have been put to use for research on inclusive MT.

**Screening questions.** As the survey required judging English→Italian translations, only participants with high competence of both Italian (C1 or higher) and English (B2 or higher) were eligible to take part in the survey. Accordingly, screening questions aimed to verify such language skills were placed at the beginning of the survey. Of the 101 received responses, 98 were from eligible participants and thus included in our analysis. The screening questions also excluded participants under 18.

**Sociodemographic information.** To gain sociodemographic information about our participant pools, the survey consisted of a short section asking for background information (e.g., educational level and field), as well as age and self-reported gender information. Overall, our pool of participants was quite homogeneous in terms of education levels

| | |
|---|---|
| cisgender woman | 1 |
| female | 1 |
| woman | 55 |
| cis male | 1 |
| male | 2 |
| man | 29 |
| lad | 1 |
| non-binary transgender | 1 |
| trans man | 1 |
| I don't define myself | 1 |
| – | 5 |

Table 6: Open responses to the question: *How do you identify?*

(i.e., the majority of respondents had a master's degree) and age (with 24-35 being most represented age range). This was expected, because of the channels through which the survey was distributed, but especially in light of the high English competence required to take part in the survey. In terms of gender, the breakdown in Figure 6 shows an higher representation of the feminine segment of the population in the survey. Since participation to the survey was voluntary, it might have attracted individuals more interested in the topic.

We do not consider this homogeneity as a limitation per se. Rather, it allowed us to gauge the opinions of relevant, interested stakeholders, which are mostly affected by discriminatory language.

**Linguistic acceptability.** The pairs of source English sentences aligned with an Italian GT and NT were created by a professional linguist with prior experience on gender-inclusive language. The original source and GT parallel sentences were retrieved from the administrative/legislative domain of EU multilingual documents. The linguist then created the second NT.

From a methodological perspective, we decided to pair the GT and NT alternatives so to allow for a fixed comparative term. Otherwise, different judgments of different NT translations alone would have not allowed for the isolation of gender-related factors from other aspects of the translations that could have influenced participants' perception of acceptability.

Though not shown in the paper due to space constrains, for each example sentence in the survey the participants were directed to follow up questions, so to motivate their choice and provide more insights on the limits of the offered NT (see Figure 2). Overall, in this section, a total of 7 example translation were shown. Additionally, for 3 source English sentences, participants were asked to pick

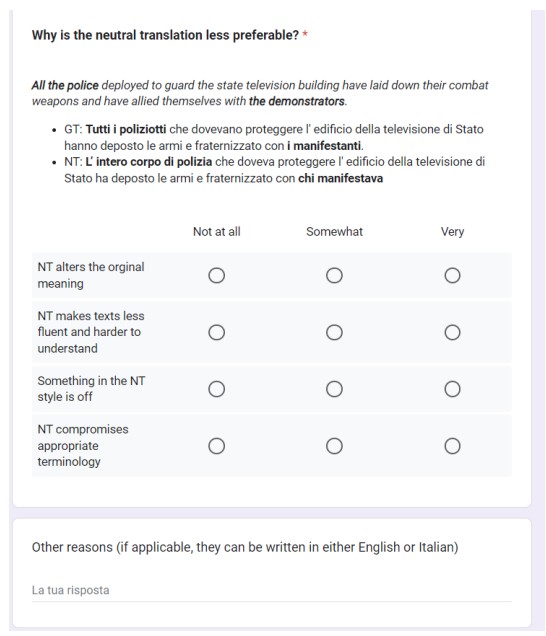

Figure 2: Questionnaire: follow-up questions on linguistic acceptability.

a preferred neutral translation from 4 different options.

**Use and attitude.** The last portion of the survey directly investigates users' attitude and perception toward gender-neutral language. For instance, in Figure 3, we attest that participants seem inclined to sacrifice brevity in favor of neutrality. Note that, since these questions focus on gender-neutral language – rather than translation – they conceptually preceded the section on linguistic acceptability. However, we placed them afterwards, so as to avoid influencing participants' responses on gender-neutral translations.

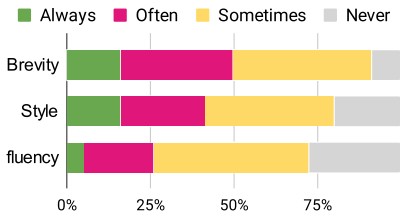

Figure 3: Willingness to sacrifice different communicative aspects to ensure neutrality.

## B    GeNTE corpus details

### B.1    Data editing report

In our data editing process, we performed two types of interventions: *(i)* editing which was functional to the creation and the optimal use of the corpus

(motivation described in §4.1); *(ii)* editing aimed to improve the overall quality of corpus data.

Functional interventions *(i)* include two procedures: (A) the editing of source and reference sentences so as to have them only include referents that require the same type of either neutral or gendered forms (203 entries total); (B) the duplication of gendered entries, in which, then, the gendered words were replaced with equivalents of the opposite gender – thus we produced 126 masculine entries and 247 feminine ones (373 entries total). Some of the entries underwent both procedures. These procedures were performed on a total of 576 entries.

With the second type of interventions *(ii)* we improved the quality of the corpus data. We did so by correcting translation errors in the original references and removing extra elements in both source and reference sentences. For example, from the source sentence "EN ) I would like, in particular, to thank Mrs Van den Burg, a Dutch Social Democrat who worked particularly hard on Article 25." we removed the segment "EN )". We performed these corrections on a total of 89 corpus entries. Moreover, to improve the variability within the data, we replaced the most frequent noun which entailed a gendered translation, *rapporteur*, with other terms from the institutional/administrative linguistic domain (e.g., *spokesperson, delegate, deputy*). We performed this operation on 70 corpus entries.

Overall, we edited 314 original source sentences and 393 original reference sentences.

### B.2 Challenges in the creation of gender-neutral references.

From a qualitative perspective, two main type of challenges were identified in the creation of the neutral references:

**Articles**: in 11 instances, the translators produced partial neutralizations, as they overlooked masculine articles. This possibly suggests that, in Italian, articles may be perceived as secondary in expressing gender compared to nouns, adjectives, and verbs, even by native speakers. Regardless, all errors were spotted by the linguist and corrected.

**Lexical gender**: translators were unable to produce a neutralization for 4 instances of lexically gendered nouns such as '*sorella*' (sister) and '*figlia*' (daughter). Such cases all concerned the creation of neutral references for the SET-G – which served the purpose of the contrastive evaluation (Sec. 5) –

but were particularly challenging as they required the use of neutral strategies for unequivocally gendered terms.

Less problematic and systematic difficulties involved specific terms which the translators struggled with, such as '*deputato*' (deputy). This is possibly due to the fact that some domain-specific terms and their translations, like 'deputy-deputato', are established and rooted in the language to the point where producing a gender-neutral translation is counter-intuitive and challenging. In all cases, the linguist intervened and proposed a solution e.g., '*persona deputata*' (lit. deputed person).

### B.3 Linguistic diversity in GeNTE's gender-neutral references

Table 7 reports our evaluation of the linguistic variability within the references of the COMMON-SET. To perform such evaluation we computed BLEU scores matching every reference of each entry with the other two references of that entry. The scores show how there is a noticeable variability, which is attributable to the different neutralization strategies employed by the three translators. On one hand, the rather high BLEU scores indicate that the references are very similar – as expected, since they share all the original content of the gendered reference, except for the gender-related terms. On the other hand, their distance from a score of 100 BLEU points indicates that they are not perfectly identical. The variability appears coherent among the sets: the scores of the neutral references evaluated against other neutral references are especially similar in COMMON-SET-N, where the highest and the lowest scores differ by less than 1 BLEU point, possibly indicating that the neutralization strategies employed in this set were indeed different, but had very similar impact on the original sentences.

## C Classifier Training

### C.1 Generation of Synthetic Training Data

**Seed words.** The generation process began with the creation of ∼200 unique triples of seed words (e.g., *il personale impiegato* [neutral] - *l'impiegato* [masculine] - *l'impiegata* [feminine]). Half of them were sourced from Europarl training data by means of keyword extraction,[19] the other 100 were instead created from scratch. We then manually augmented this initial list of triplets by re-generating them with various inflectional morphology, which is relevant

---

[19] https://pypi.org/project/rake-nltk/

| COMMON-SET-G | | | | |
|---|---|---|---|---|
| ↓ REF \| CAND → | Reference 1 | Reference 2 | Reference 3 | REF-G |
| Reference 1 | - | 75.14 | 77.65 | 74.14 |
| Reference 2 | 75.14 | - | 75.09 | 72.08 |
| Reference 3 | 77.59 | 75.03 | - | 74.89 |
| REF-G | 74.04 | 71.98 | 74.82 | - |
| COMMON-SET-N | | | | |
| ↓ REF \| CAND → | Reference 1 | Reference 2 | Reference 3 | REF-G |
| Reference 1 | - | 76.88 | 76.27 | 75.89 |
| Reference 2 | 76.91 | - | 76.15 | 73.36 |
| Reference 3 | 76.28 | 76.14 | - | 73.02 |
| REF-G | 75.78 | 73.26 | 72.92 | - |

Table 7: BLEU scores representing the linguistic variability in COMMON-SET's references.

to distinguish for the task of neutralization (e.g., make them plural, use indefinite article etc.). Accordingly, we obtained ∼800 triplets of seedwords.

**Generation: first round.** We prompted the `gpt-3.5-turbo` model from the GPT LLM family[20] to generate triplets of sentences given the triplets of seed words. We used a few-shot approach with given examples of the task to be performed (see Figure 4.). We access the model via OpenAI paid API and setting a temperature of 0.5.[21] In total, approximately 60,000 sentences were generated. A random sample of 100 sentences was manually inspected, revealing that noise was very low (10%), but that the sentences exhibited a simple structure, consistently placing the subject at the beginning.

**Generation: second round.** To enhance the quality and textual context of the generated sentences, a second round of generation was performed using a lower temperature of 0.3 (see Figure 5). Each triplet of sentences was rewritten multiple times in different forms. This process resulted in the generation of approximately 320,000 sentences, which had a higher occurrence of incorrect alternatives for the seedwords, estimated to be around 40% based on the inspection of 100 randomly selected sentences. The final synthetic corpus consists of approximately 380,000 sentences, featuring varied sentence structures. Specifically, one-third of the sentences contain a masculine seedword, another third contain a feminine seedword, and the remaining third contain a neutral seedword.

Overall, a cost of $13.12 USD was estimated.

### C.2 Training Setup

We trained the parameters of both the linear layer and UmBERTo on the classification task for 2 epoch, with learning rate of 5e-5, batch size of 64 and maximum sequence length of 64, on a p3.2xlarge instance on AWS (featuring one NVIDIA V100 GPU). The code for finetuning relies on Huggingface transformers library (Wolf et al., 2020).

---

[20]https://platform.openai.com/docs/models/gpt-3-5
[21]https://platform.openai.com/docs/api-reference

Figure 4: **Prompt template** for the generation of triplet of sentences from (NEUT/FEM/MASC) seed words.

Figure 5: **Prompt Template** for the rewriting of triplet of (NEUT/FEM/MASC) seed sentences.