# OpenReview forum: "Hi Guys or Hi Folks? Benchmarking Gender-Neutral Machine Translation with the GeNTE Corpus"
_EMNLP/2023/Conference — EMNLP 2023 Main_

### Official Review · Reviewer_sWdu · 2023-07-30

**Soundness:** 4

**Excitement:**

4: Strong: This paper deepens the understanding of some phenomenon or lowers the barriers to an existing research direction.

**Paper Topic And Main Contributions:**

The paper describes the creation and analysis of GeNTE a `natural, bilingual test set for gender-neutral translation, whose creation was informed by a survey on the perception and use of neutral language.'     The authors provide an accurate summary of their contributions:
(1) A study on the feasibility of neutral translation, (2) creation of GeNTE,1 the first natural, parallel corpus designed to test MT systems’ ability to generate neutral translations; (3) A comprehensive analysis of the (un)suitability of existing automatic metrics to evaluate neutral translation.

**Questions For The Authors:**

Did you consider revisiting the survey done in section 3 but asking respondents to judge the created data itself?


**Reasons To Accept:**

Very useful test set.   Excellent introduction and motivation for the problem of gender neutral translation.  The background and survey are really nicely presented, as is the small-scale study surveying translator's preferences wrt neutral translations.    Overall,  a very strong case is made for working on the problem and for the GeNTE corpus.   The work is likely to be influential.

**Reasons To Reject:**

As the authors note in their limitations section,  the collection focuses on English->Italian translation,   and so the data and classifier models can't be used directly for other target languages.    Personally,  I'm ok with this,  as I think these subtle issues are best analyzed for specific target languages.   While this is a limitation, the paper also provides a starting framework for approaching other target languages.

I may have missed this in the discussion,   and I did not read the appendices,  but apart from a summary statement (line 366 - `the linguist supported the translators throughout the process and finally checked all the neutralizations')  there isn't much discussion about the quality of the data set that was produced.    It would be interesting to know whether there were challenges with inter-annotator agreement, i.e. whether the linguist needed to intervene often,  or how often the annotators 'failed to neutralise' (Table 2, iii-F,  line 395).


**Reproducibility:**

4: Could mostly reproduce the results, but there may be some variation because of sample variance or minor variations in their interpretation of the protocol or method.

**Reviewer Confidence:**

3: Pretty sure, but there's a chance I missed something. Although I have a good feel for this area in general, I did not carefully check the paper's details, e.g., the math, experimental design, or novelty.

**Typos Grammar Style And Presentation Improvements:**

I'm not sure if this is intentional,  but I'd suggest settling on a single spelling of 'ambiguous'.

---

> ### Author Rebuttal · Authors · 2023-08-29
>
> **QUALITY OF THE DATASET & IAA**.
> As an open-ended task, translation allows for multiple acceptable alternatives rather than a single correct option. For this scenario, inter-annotator agreement does not offer a reliable indication of quality, as it risks penalizing different and yet equally good translations. To assess consistency with a less stringent approach, we instead manually analyzed all the references in the common-set and found that the translators produced identical gender-neutral references in 13.57% of the cases, while an additional 8% exhibited a high degree of similarity (e.g., the same neutral words are used, but in a different order). These statistics are positive: they show that our common-set presents a good level of variability (~80%), which is desirable to test open-ended generation tasks like MT. Also – and especially in light of the fact that the translators worked independently – the ~20% of identical/similar neutralizations suggests that neutralizing translation is a challenging but feasible task.
>
> To ensure the good quality of the final data, we also pipelined their creation methodology: i) first, the linguist annotated the gendered words of interest in each sentence, ii) following detailed instructions, the professional translators worked to neutralize the already identified gendered expressions, and finally, iii) the linguist revised the neutral translations produced. From a qualitative perspective, two main type of challenges were identified in the creation of the neutral references:
>
>  **(A) Articles**: in 11 instances, the translators produced partial neutralizations, as they overlooked masculine articles. This possibly suggests that, in Italian, articles may be perceived as secondary in expressing gender compared to nouns, adjectives, and verbs, even by native speakers. Regardless, all errors were spotted by the linguist and corrected.
>
> **(B) Lexical gender**: translators were unable to produce a neutralization for 4 instances of lexically gendered nouns such as ‘sorella’ (sister) and ‘figlia’ (daughter). Such cases all concerned the creation of neutral references for the SET-G – which served the purpose of the contrastive evaluation (sec. 5.1) – but were particularly challenging as they required the use of neutral strategies for unequivocally gendered terms.
>
> We appreciate the reviewer’s interest in these aspects of the creation of the corpus and we will consider including this discussion and data in the camera-ready version of the paper.
>
> **ADDITIONAL SURVEY TO JUDGE THE CREATED DATA**.
> We deliberately placed the survey at the outset of our research, so as to gather valuable insights head-on and provide direction for the subsequent development of the GeNTE corpus, drawing on valuable feedback from end-users. Given such feedback and insights, we then entrusted the expert linguist and professional translators to successfully implement them in the creation of the corpus. While we believe that the employed methodology represents the soundest and most natural sequence of steps to ensure the good quality of our data, closing the whole process with a second round of end-users’ feedback would indeed be interesting and could further validate our corpus and task. We thank the reviewer for the suggestion, which we will consider for future work.

---

### Official Review · Reviewer_VboY · 2023-08-01

**Typos Grammar Style And Presentation Improvements:** "ambiguos" --> ambiguous
**Soundness:** 3

**Excitement:**

3: Ambivalent: It has merits (e.g., it reports state-of-the-art results, the idea is nice), but there are key weaknesses (e.g., it describes incremental work), and it can significantly benefit from another round of revision. However, I won't object to accepting it if my co-reviewers champion it.

**Missing References:**

- A newer Italian Language model: BureauBERTo. https://colinglab.fileli.unipi.it/wp-content/uploads/2023/04/Ital_IA_2023_BureauBERTo.pdf
- References to other gender datasets such as GAP (https://aclanthology.org/Q18-1042.pdf)
- More on gender bias in translation: https://aclanthology.org/2021.tacl-1.51.pdf
- Gender suffix in German: https://aclanthology.org/2022.konvens-1.10.pdf
- "In some sense, a well-biased translator is a well-performing translator that reflects
the inter-cultural difference." from https://aclanthology.org/W19-3824.pdf

**Paper Topic And Main Contributions:**

This paper introduces GeNTE, an English-->Italian test set balanced between Neutral (N) and Gendered (G) English nouns, to evaluate faithfulness when translating gender-neutral nouns from English to Italian, a language in which masculine is the default gender.
In preparation for the data and evaluations, they conducted a survey regarding N vs. G Italian translations in formal and informal settings.
They also evaluate reference-based n-gram and neural evaluation metrics on gender-neutral MT and suggest a reference-free accuracy evaluation on such translations.

**Reasons To Accept:**

The paper focuses on a special case of gender bias across translations -- when the source gender is ambiguous or ungiven whereas the translation defaults to specific genders.

An open-source survey is conducted on 100 participants to assess how acceptable and preferable are the gender-neutral translations for the general audience.
The demographical details of the participants are illustrated in the Appendix.

The paper points out that 97.2% of segments collected from Europarl are biased toward gendered references in Italian.
Thus, GeNTE devotes the effort to manually correcting referent gendered and neutral sentences.

**Reasons To Reject:**

Some major concerns:
- Gender neutrality assumption in the source language. Though some nouns are lexically gender-neutral in English, document context could render these nouns gender-specific. Relevant questions: 1) is context taken into consideration while filtering gender-neutral English sources? 2) did the survey include questions regarding source gender neutrality?
- Over-neutralization. For specific nouns as in example C of Table 1, much less participants prefer the gender-neural alternative in Italian. Is it necessary for these target translations to be gender-neutral? Also in Table 2, why would we need to neutralize ii and iii for translations if the source English is already gendered?
- On multiple N references. How frequently do the three translations overlap or are identical? Have you considered multi-reference evaluations?
- Sentence vs. document level. How are the sentence-level N-vs-G tags migrated, are they inherited directly from the document level?
- Problematic reference-free gender-neutral MT evaluation. The binary N-G classification accuracy only evaluates the scenario when a gender-neutral noun is translated into Italian. It does not provide any information on the overall translation quality of the documents/sentences thus not linearly dependent and cannot be compared proportionally to standard MT metrics, BLEU, TER, and METEOR. Unfortunately, the much higher "classifier" number in Table 5 does not lead to the conclusion that the reference-free evaluation is "promising" (line 625).

Minor clarifications are necessary:
- There is no concrete accuracy assessment on the two rounds of GPT-generated training data;
- "Grammatical gender languages" is a confusing term in this paper. Only concepts that have M-F-N cognates are examined in this paper, but not other nouns that have inherent gender in Italian, such as desks, beds, etc.

**Reproducibility:**

3: Could reproduce the results with some difficulty. The settings of parameters are underspecified or subjectively determined; the training/evaluation data are not widely available.

**Reviewer Confidence:**

4: Quite sure. I tried to check the important points carefully. It's unlikely, though conceivable, that I missed something that should affect my ratings.

---

> ### Author Rebuttal · Authors · 2023-08-29
>
> We thank the reviewer for the valuable comments, which give us the chance to further elaborate on the goals of our work as well as to improve on the final version of the paper. We are glad to clarify all of the raised concerns – especially those that question the soundness of our work and present overly stringent justifications for paper rejection. We believe that the major problems reported in the review do not subsist and hope to solve remaining doubts surrounding our submission.
>
>
>
> **GENDER NEUTRALITY ASSUMPTION IN THE SOURCE LANGUAGE**.
>
> **(1) Context.** While the data extraction was carried out automatically, all extracted segments were manually checked, selected and annotated as either G (gendered) or N (neutral) by a professional linguist. In doing so, context was definitely taken into account. For instance, “They are police officers…” is to be annotated as N (to be neutralized in translation, since no specific information is given in the source sentence), whereas a sentence like “She is a police officer…” is considered G (to be rendered as gendered in translation, given the presence of the disambiguating gendered cue ‘she’, even though ‘police officer’ is lexically gender-neutral). Further examples for the word ‘colleague’ are also given in Table 2. We stress that the identification of contextual information as gender-specific cues (e.g., ‘she’, but also ‘Mr./Madam’ etc.) was a key portion of the selection/annotation process, as – perhaps too swiftly – mentioned at lines 297-300. Granted the additional page in the camera-ready version of the paper, we will include a more detailed description of this context-aware selection process, which was carried out at the sentence level. Given i) the novelty of the task represented by our corpus, and ii) the fact that current MT landscape predominantly consists of SOTA systems that perform translation at the sentence level, we decided not to account for a larger, document-level context: it would have added additional complexity to an already challenging – and yet to be truly explored – task.
> **(2) Survey.** To the best of our understanding of the question, the survey did not include *direct* questions on source gender-neutrality (i.e., is this English sentence neutral?). The example sentences included in the survey were selected and created by an expert on the topic of gender-inclusive language to inform i) when and ii) how neutralization should be performed in translation (for more details see also Appendix A). Our approach was therefore *indirect*: by asking participants if they found the neutral translation acceptable compared to a gendered one, we also indirectly assessed the ‘neutrality’ of the source sentence (i.e., if the participants considered it an example that required a neutral translation).
>
>
> **OVER-NEUTRALIZATION**.
> Our work is motivated by the intent to avoid using gendered terms in translation when unnecessary. Such is the case for a source sentence like example C in Table 1. Lower scores for the Neutral Translation here (though we stress that the majority of respondents found the Gendered Translation and the Neutral Translation to be equally acceptable) are to be attributed to the type of target neutralization strategy used, which – as described at lines 235-237 – was found by some respondents to sacrifice adequacy for neutralization. Indeed, we were aware of potential trade-offs to consider in performing neutral translation, and the survey served also to investigate them. Accordingly, it is not that source example C *should not* require a neutral translation; rather, the survey informed us that a different linguistic strategy for neutralization should be employed to better preserve the meaning of the source sentence while still ensuring inclusivity in the target language. In fact, example C is given as a negative example not to follow in the guidelines provided to the translators who created GeNTE’s neutral references (see the supplementary materials, example 7).
> Examples II and III in Table 2, instead, are cases from the G-SET of the corpus, i.e., sentences that should *not* be neutralized in translation. As described in Sec 4.1 and 5.1., the creation of alternative neutral references for II and III serves the purposes of the contrastive, reference-based evaluation protocol. Please refer to our answer below for more on this point (see *REFERENCE-FREE EVALUATION*). We thus reject the argument that they represent wrongfully created instances of over-neutralization.
>
>
> **MULTIPLE NEUTRAL REFERENCES**.
> Based on a manual analysis of the common-set, the translators produced three identical gender-neutral references in 13.57% of the cases, while an additional 8% of translations exhibited a high degree of similarity (e.g., the same neutral words are used, but in a different order). These statistics are positive: they show that our common-set exhibits a good level of variability (~80%), which is desirable to test open-ended generation tasks like MT. Also – and especially in light of the fact that the translators worked independently – the ~20% of identical/similar neutralizations suggests that neutralizing translation is a challenging but feasible task.
> We did not perform a multi-reference evaluation since each entry in the common-set contains 1 gendered reference only vs. 3 neutral references. This imbalance would have compromised the soundness of our experiments with the risk of skewing the results. Granted the extra-space of the camera-ready, we will include the above statistics and these considerations.
>
> **SENTENCE VS DOCUMENT LEVEL GENDER ANNOTATION**.
> The Europarl corpus consists of parliamentary speeches annotated with the gender of the associated speaker [1]. For GeNTE, however, we did not rely on such document-level annotations, as our corpus represents several types of referents beside the speaker to ensure more variability. As described in our previous answer (see *Gender neutrality assumption in the source language*), the N vs G tags were carefully assigned to each sentence by a professional linguist who accounted for the sentence-level context.
>
> [1] Vanmassenhove et al. (2018). [Europarl Datasets with Demographic Speaker Information](https://aclanthology.org/2018.eamt-main.59/).
>
> **REFERENCE-FREE GENDER-NEUTRAL MT EVALUATION**.  Our reference-free evaluation metric (i.e. classifier) was purposefully designed to focus only on the assessment of gender-neutral translation and is not intended to also provide information on overall translation quality.  Although integrating these two aspects in a single metric could indeed prove valuable, this comes with the non-negligible challenge posed by the combination of different ‘scores’ that measure different aspects. For this reason -  and in line with current practices in MT evaluation (e.g. those based on test suites) - keeping the two aspects separated was deemed preferable to present a first, pinpointed assessment of novel and specific gender-related aspects.
> In Table 5 we compare the results obtained from the classifier with those obtained from a reference-based evaluation protocol that relies on standard MT metrics. This protocol, however, does not exploit those metrics to measure overall translation quality but to make them informative about gender. As described in Sec. 5.1., the proposed contrastive approach – computed on the double set of GeNTE references (Neutral vs Gendered) – assumes that a NEUTRAL output translation should obtain higher scores on the N-reference compared to the G-reference, and vice versa for a GENDERED output translation. In this way, when applied at the sentence level, the contrastive reference-based evaluation protocol that relies on BLEU, TER and METEOR functions as a classifier (e.g. higher BLEU on the N-reference wrt. the G-reference ⇒ output sentence classified as NEUTRAL). Accordingly, in Table 5 we can i) directly and soundly compare the results of the reference-based evaluation with the reference-free one, and ii) show that the latter is more accurate in correctly classifying NEUTRAL vs GENDERED output translations, thus emerging as a first, promising step toward enabling the assessment of inclusivity in MT.
>
> **ASSESSMENT OF GPT-GENERATED DATA**.
> The description of the generation of GPT-data is reported in Appendix C.1., where we also explain how the data have been assessed by means of a manual analysis on a random sample of 100 triplets (i.e. 300 sentences) for each generation round.
>
>
> **MISSING REFERENCES**.
> We thank the reviewer for pointing out the study on “Gender suffix in German” by Wagner & Zarrieß (n.4).  It is indeed relevant to our work, and we will make sure to include it in the final version of our paper. Concerning the other references, however, we would like to stress that to consider them *missing* is partially inaccurate and seems overly penalizing. In fact, the work by Savoldi et al., (2021 - n.3) and Cho et al., (2019 -n.5) are both already referenced in the paper – respectively, at lines 37 and 131. Also, paper n.2 is not highly relevant, since it refers to the English, monolingual GAP corpus for coreference resolution on binary gender, and paper n.1 was published on May 29th, 2023, and it is thus concurrent to our submission.

---

### Official Review · Reviewer_qfqY · 2023-08-03

**Soundness:** 4

**Excitement:**

4: Strong: This paper deepens the understanding of some phenomenon or lowers the barriers to an existing research direction.

**Paper Topic And Main Contributions:**

This paper addresses the challenge of correctly handling gender in machine translation, and particularly making sure that gender-neutral language is used when appropriate. To this end, they start by conducting a study to understand how people react to language that has been changed to be gender neutral when the actual gender of the referent is unknown or irrelevant, and they find that people largely preferred the gender-neutral edited language, particularly in formal settings.
They then created a test set of gender neutral and gendered language for EN-IT, as well as a test bed of MT outputs, some of which were post-edited to use correct gender-neutral language, and experimented with the goodness of existing reference-based metrics for evaluating the correctness of whether translation outputs correctly output gendered or gender-neutral forms. They find that existing metrics have little utility for this task, particularly neural metrics.
Finally, they proposed a classifier for predicting whether a sentence contains gender-neutral or gendered language, and find that it vastly outperforms generic MT metrics not intended for this task.

**Questions For The Authors:**

How would you integrate the proposed evaluation metric with other measures that also attempt to capture other aspects of translation goodness, i.e., fluency/adequacy?

Can you supply judgements of the translators who created the eval set? How often did the translators create gender-neutral translations they were unhappy with? I have some concern that without being very attentive to this concern, we could be at risk of imposing (largely non-inflecting) English-based concepts of language onto other languages where gender has extensive grammatical function, where in English it's largely semantic. A more robust description of the edited gender-neutral translations and their naturalness and appropriateness for native speakers would help allay this concern.

**Reasons To Accept:**

The main strengths of this paper are:
1. It presents the first test set and benchmark for measuring how well MT systems do at appropriately generating gender-neutral language. This will be a valuable resource for understanding this dimension of the translation task.
2. Before launching the evaluation task, they ran a study assessing the naturalness of language that had been edited to be gender neutral and how speakers reacted to the edited language. This was an important step in justifying the subsequent work, showing that people typically appreciated the neutral language, and that it was not overly contorted.
3. They show that conventional MT metrics don't capture this dimension of the MT task well, and that other methods are needed.
4. They train a classifier that is effective at predicting the use of gender-neutral language.


**Reasons To Reject:**

The main weaknesses are:
1. The methods used to collect the test set data may not have been ideal. E.g., they used a regex to find instances of gender-unambiguous language, including gendered pronouns. However, in English, gendered pronouns are sometimes used in what should be gender-neutral circumstances ("To each his own").
2. They don't supply statistics about how difficult/natural/realistic it was to create the eval set. I.e., we don't see how often the translators tasked with creating the eval set had difficulty in creating natural gender-neutral edited language, only that one example was so difficult to make gender neutral that only one of the translators attempted it.

Minor:
1. To create the benchmark set, they post-edited MT outputs to use more gender-neutral language, thus not a fully realistic scenario. However, they discuss this in the limitations section, and it seems like a warranted compromise, given the extreme bias of MT systems to use gendered language.
2. The reference-free gender classifier doesn’t account for how good the translation is overall, and thus would not penalize translations that are incorrect other than their handling of gender.

**Reproducibility:**

3: Could reproduce the results with some difficulty. The settings of parameters are underspecified or subjectively determined; the training/evaluation data are not widely available.

**Reviewer Confidence:**

4: Quite sure. I tried to check the important points carefully. It's unlikely, though conceivable, that I missed something that should affect my ratings.

---

> ### Author Rebuttal · Authors · 2023-08-29
>
> **METHOD TO COLLECT TEST DATA**.
> The regular expressions were used to create initial pools of – automatically extracted – candidate sentences which contained our phenomena of interest. The actual sentence selection, however, was carried out manually by a professional linguist who carefully evaluated and annotated each sentence. During this process, the sentence context was indeed used to differentiate between the use of gendered words as either masculine generics (e.g., “If this directive were adopted, it would then be up to the accused employer to prove *his* innocence…” – identified as N) or as informative of a referent’s gender (e.g. “I should like to mention that the President of the Republic will be accompanied, as is usually the case, by *his* delegation” – identified as G). The differentiation of such cases represents a key aspect of the GeNTE creation process (as already mentioned at lines 305-310), which we will discuss in higher detail in the camera-ready version of the paper.
>
>
> **DIFFICULTIES IN CREATING NATURAL GENDER-NEUTRAL REFERENCES**.
> We did not add this information due to space reasons. While producing the gender-neutral references, there were four instances where two of the translators reported difficulties in neutralizing lexically gendered terms, i.e. ‘figlia’ (daughter) and ‘sorella’ (sister), as shown in Table 2, example *iii*. It should be noted that such words are not supposed to be translated as gender-neutral and we only tasked the translators with doing so in order to produce the neutral references necessary to enable the reference-based contrastive evaluation (Sec. 5.1.) on the double set of G vs N references. Less problematic difficulties involved specific terms which the translators struggled with, such as ‘deputato’ (deputy). This is possibly due to the fact that some domain-specific terms and their translations, like ‘deputy-deputato’, are established and rooted in the language to the point where producing a gender-neutral translation is counterintuitive and challenging. In all cases, the linguist intervened and proposed a solution (e.g., ‘persona deputata’, lit. ‘deputed person’). Moreover, there was one common pattern of errors: in 11 instances they overlooked masculine articles, thus only producing partial and incomplete neutralizations. This possibly suggests that, in Italian, articles and prepositions could be perceived as secondary in expressing gender compared to nouns, adjectives, and verbs, even by native speakers. Regardless, the linguist highlighted such errors and the translators were able to produce a different neutralization.
> As these examples show, English linguistic strategies for gender-neutralization cannot – and were not – directly transposed and applied to the Italian language. Rather, in the creation process of GeNTE we had a strong focus on the features of grammatical gender languages as to avoid being primed on English strategies. For this reason, the creation of GeNTE was (1) motivated and guided by recent literature on gender-neutral translation in Italian (see [1] and [2]); (2) informed by the survey described in Sec. 3, which analyzes the native speakers’ perception of gender-neutral translation; (3) entrusted to professional translators, native speakers of Italian. Overall, given the small number of problematic instances mentioned-above, we believe that that neutralization task is challenging but feasible.
>
> [1] Lardelli, Manuel & Gromann, Dagmar. (2023). [Gender-Fair (Machine) Translation](https://www.researchgate.net/publication/369948882_Gender-Fair_Machine_Translation).
>
> [2] Piergentili et al. (2023). [From inclusive language to gender-neutral machine translation](https://arxiv.org/abs/2301.10075).
>
> **(REFERENCE-FREE) GENDER ASSESSMENT AND OVERALL TRANSLATION QUALITY**.
> This work maintains a distinction between the evaluation of gender neutrality (as a discrete binary measure) and other continuous measures of overall translation quality (i.e., fluency/adequacy). Hereby, such a distinction was deemed preferable to present a first, pinpointed assessment of novel and specific gender-related aspects. However, integrating these aspects could prove valuable, yielding comprehensive and multifaceted evaluations. This comes with the non-negligible challenge posed by the combination of different ‘scores’ that measure different aspects. To achieve this in future work, in the wake of studies such as [1] and [2], instruction-tuned models could be used to develop multidimensional evaluations based on detailed schemes with the potential to inform scalar quality scores. Also, upon the availability of the created synthetic gender-neutral data (Sec. 5.2), neural metrics and QE could be retrained and adapted for evaluating gender-neutral translation, similarly to [3].
>
> [1] Fernandes et al., (2023). [The Devil is in the Errors: Leveraging Large Language Models for Fine-grained Machine Translation Evaluation](https://arxiv.org/abs/2308.07286).
>
> [2] Lai et al., (2023).  [Multidimensional Evaluation for Text Style Transfer Using ChatGPT](https://arxiv.org/abs/2304.13462).
>
> [3] Sharami et al., (2023). [Tailoring Domain Adaptation for Machine Translation Quality Estimation](https://arxiv.org/abs/2304.08891).

---

### Official Review · Reviewer_yjUc · 2023-08-22

**Soundness:** 4

**Excitement:**

4: Strong: This paper deepens the understanding of some phenomenon or lowers the barriers to an existing research direction.

**Paper Topic And Main Contributions:**

In this paper, the authors investigate gender-neutral translation for inclusive Machine Translation, English to Italian as the testbed. The authors provide a thorough overview of existing research on gender-neutral/gender-aware translation and identify the research gap clearly. Beginning with a human study to gauge the feasibility of gender-neutral translation, the authors create a parallel corpus containing both gender-ambiguous source sentences (where neutralization may be preferred) and gender-unambiguous source sentences (where the appropriate gender is desired in the target language) to foster research in this field. Finally, the authors study the effectiveness of traditional reference-based evaluation metrics and their shortcomings, and propose an evaluation metric to address those gaps.

**Questions For The Authors:**

A. Can you quantify how post-editing the neutral translations that are generated from Amazon and DeepL affect their naturalness?

**Reasons To Accept:**

1. A thorough review of existing research in neutralization and gendered translations in MT.
2. A carefully constructed parallel corpus containing sentences that facilitate evaluation in both gender-ambiguous and gender-neutral scenarios.
3. A review of how existing reference-based metrics fare in detecting the quality of neutral translations, and their particular shortcomings. For example, the fact that lower frequencies of neutral expressions affect the neural metrics more than the n-gram overlap-based ones, and the final finding that even n-gram metrics are not fully reliable at the sentence level.
4. A new, reference-free evaluation metric that accounts for the shortcomings of the reference-based ones.

**Reasons To Reject:**

The reasons for a new metric are well-motivated, but it is unclear how good the new metric is at gauging the quality of the resulting translation, aside from detecting whether a translation is gendered or not. This new classification metric perhaps needs to be validated with the human study to make sure it's measuring the quality of translations as well. Because of this reason, it is unclear what it means when the authors say "the classifier outperforms" (line 614).

This is something I might have misunderstood, in which case I welcome the authors to clarify further. Although, I really appreciate the authors being honest about the thoughtful limitations section.

**Reproducibility:**

3: Could reproduce the results with some difficulty. The settings of parameters are underspecified or subjectively determined; the training/evaluation data are not widely available.

**Reviewer Confidence:**

3: Pretty sure, but there's a chance I missed something. Although I have a good feel for this area in general, I did not carefully check the paper's details, e.g., the math, experimental design, or novelty.

**Typos Grammar Style And Presentation Improvements:**

Typos -

1. Grammer: "Ambiguos" -> ambiguous [Line 189, Line 281]

---

> ### Author Rebuttal · Authors · 2023-08-29
>
> **CLASSIFIER AND OVERALL TRANSLATION QUALITY**.
> Our reference-free evaluation metric (i.e. classifier) was purposefully designed to focus only on the assessment of gender-neutral translation. In line with current practices in MT evaluation (e.g. those based on test suites), this new metric does not aspire to also gauge overall translation quality and is rather intended to complement established metrics for generic performance. For this reason, a human-evaluation study on the matter was not conducted.
> In the paper we compare the results obtained from the classifier with those obtained from a reference-based evaluation protocol that relies on standard MT metrics (Table 5). However,  the protocol does not exploit those metrics to measure overall translation quality but to make them informative about gender. Specifically, it proposes a contrastive method computed on the double set of GeNTE references (described in Sec. 5.1) that, when applied at the sentence level, relies on BLEU, TER, and METEOR to function as a classifier of neutral vs. gendered output translations. As such, the comparison between the two protocols is possible and shows that the classifier outperforms the reference-based method in correctly classifying neutral vs. gendered output translations.
>
>
> **NATURALNESS OF POST-EDITED MT OUTPUTS**.
> We confined the post-editing process to gender-related words only, so as to avoid altering the ‘naturalness’ (intended as the original level of adequacy/fluency) of the MT-generated outputs. Further small adjustments were only occasionally needed to preserve the (original) grammaticality of the sentence: e.g., to ensure number agreement as in “the neighbors are” [i vicini sono - G/Plur.] vs. “the neighborhood is” [il vicinato è - N/Sing.]. On average, 12% of the words present in the systems’ output were substituted through post-editing, suggesting that these edits have a minimal and circumscribed impact on the original sentences.

---

### Meta-Review · Area_Chair_8LiR · 2023-09-07

**Recommendation:** 5

**Metareview:**

This paper presents a valuable resource to mitigate gender bias in machine translation in the form of GeNTE, an English to Italian test set for gender-neutral translation. It also investigates existing reference-based metrics for evaluating the quality of neutral translations and proposes a reference-free method that addresses the formers’ shortcomings. The strengths of the paper lie in its novelty and thoroughness, as indicated by its Soundness (three 4s, one 3) and Excitement (three 4s, one 3) scores. Multiple reviewers praised the clarity of the introduction, the motivation (particularly the human subjects survey on attitudes toward gender-neutral language), and discussion of limitations, though individual concerns were raised in regards to methodology, e.g. the use of regexes in data collection, the assumption that gender-neutral lexical items remain neutral in context, etc. The authors nevertheless addressed each of these concerns in their rebuttals to reviewer satisfaction.

In light of this, only minor revisions, addressing reviewers’ comments and questions, need to be made to ensure this paper is camera ready.

---

### Decision · Program_Chairs · 2023-10-07

**Decision:**

Accept-Main

**Comment:**

This paper presents a valuable resource to mitigate gender bias in machine translation in the form of GeNTE, an English to Italian test set for gender-neutral translation. It also investigates existing reference-based metrics for evaluating the quality of neutral translations and proposes a reference-free method that addresses the formers’ shortcomings. The strengths of the paper lie in its novelty and thoroughness, as indicated by its Soundness (three 4s, one 3) and Excitement (three 4s, one 3) scores. Multiple reviewers praised the clarity of the introduction, the motivation (particularly the human subjects survey on attitudes toward gender-neutral language), and discussion of limitations, though individual concerns were raised in regards to methodology, e.g. the use of regexes in data collection, the assumption that gender-neutral lexical items remain neutral in context, etc. The authors nevertheless addressed each of these concerns in their rebuttals to reviewer satisfaction.

In light of this, only minor revisions, addressing reviewers’ comments and questions, need to be made to ensure this paper is camera ready.